# Adulis and the transshipment of baboons during classical antiquity

Franziska Grathwol[1], Christian Roos[2], Dietmar Zinner[3,4,5], Benjamin Hume[1,6], Stéphanie M Porcier[7], Didier Berthet[8], Jacques Cuisin[9], Stefan Merker[10], Claudio Ottoni[11], Wim Van Neer[12,13], Nathaniel J Dominy[14,15], Gisela H Kopp[1,15,16,17]*

[1]Department of Biology, University of Konstanz, Konstanz, Germany; [2]Gene Bank of Primates and Primate Genetics Laboratory, German Primate Center, Leibniz Institute for Primate Research, Göttingen, Germany; [3]Cognitive Ethology Laboratory, German Primate Center, Leibniz Institute for Primate Research, Göttingen, Germany; [4]Department of Primate Cognition, Georg-August-University of Göttingen, Göttingen, Germany; [5]Leibniz-ScienceCampus Primate Cognition, Göttingen, Germany; [6]SequAna – Sequencing Analysis Core Facility, University of Konstanz, Konstanz, Germany; [7]Laboratoire CNRS ASM « Archéologie des Sociétés Méditerranéennes » (UMR 5140), Université Paul-Valéry, LabEx Archimede, Montpellier, France; [8]Musée des Confluences, Lyon, France; [9]Muséum National d'Histoire Naturelle, Paris, France; [10]Department of Zoology, State Museum of Natural History Stuttgart, Stuttgart, Germany; [11]Centre of Molecular Anthropology for Ancient DNA Studies, Department of Biology, University of Rome Tor Vergata, Rome, Italy; [12]Royal Belgian Institute of Natural Sciences, Brussels, Belgium; [13]Department of Biology, KU Leuven, Leuven, Belgium; [14]Departments of Anthropology and Biological Sciences, Dartmouth College, Hanover, United States; [15]Zukunftskolleg, University of Konstanz, Konstanz, Germany; [16]Department of Migration, Max Planck Institute of Animal Behavior, Konstanz, Germany; [17]Centre for the Advanced Study of Collective Behaviour, University of Konstanz, Konstanz, Germany

*For correspondence: gisela.kopp@uni-konstanz.de

Competing interest: The authors declare that no competing interests exist.

**Abstract** Adulis, located on the Red Sea coast in present-day Eritrea, was a bustling trading centre between the first and seventh centuries CE. Several classical geographers—Agatharchides of Cnidus, Pliny the Elder, Strabo—noted the value of Adulis to Greco-Roman Egypt, particularly as an emporium for living animals, including baboons (*Papio* spp.). Though fragmentary, these accounts predict the Adulite origins of mummified baboons in Ptolemaic catacombs, while inviting questions on the geoprovenance of older (Late Period) baboons recovered from Gabbanat el-Qurud ('Valley of the Monkeys'), Egypt. Dated to ca. 800–540 BCE, these animals could extend the antiquity of Egyptian–Adulite trade by as much as five centuries. Previously, Dominy et al. (2020) used stable isotope analysis to show that two New Kingdom specimens of *Papio hamadryas* originate from the Horn of Africa. Here, we report the complete mitochondrial genomes from a mummified baboon from Gabbanat el-Qurud and 14 museum specimens with known provenance together with published georeferenced mitochondrial sequence data. Phylogenetic assignment connects the mummified baboon to modern populations of *P. hamadryas* in Eritrea, Ethiopia, and eastern Sudan. This result, assuming geographical stability of phylogenetic clades, corroborates Greco-Roman historiographies by pointing toward present-day Eritrea, and by extension Adulis, as a source of baboons for Late Period Egyptians. It also establishes geographic continuity with baboons from the fabled Land of Punt (Dominy et al., 2020), giving weight to speculation that

Punt and Adulis were essentially the same trading centres separated by a thousand years of history.

## Editor's evaluation

This fundamental Research Advance sheds new light on the ancient baboon trade in the Red Sea. Combining ancient DNA methods from a mummified baboon with historical accounts, this work provides compelling evidence connecting the ancient Egyptian trade of baboons with the ancient port city of Adulis. The results will be of significance to a broad range of scholars interested in applying ancient DNA to improve our knowledge of historical events.

## Introduction

Adulis, on the coast of present-day Eritrea, was an important hub during the rise of cross-ocean maritime trade, connecting ships, cargoes, and ideas from Egypt, Arabia, and India (*Burstein, 2002*; *Munro-Hay, 1982*; *Seland, 2008*). Trade peaked between the fourth and seventh centuries CE, propelling the rise and expansion of the Aksumite kingdom, but its occupation history extends, at minimum, to the first millennium BCE (*Zazzaro et al., 2014*). Corroborating this archaeological record are written accounts that draw attention to the importance of Adulis as one of the foremost sources of African animals or animal products during the Hellenistic period (323–31 BCE). In *Topographia Christiana*, a sixth-century text, the Nestorian merchant Cosmas Indicopleustes recounts his own visit to Adulis in 518 CE (*Fauvelle-Aymar, 2009*; *Hatke, 2013*). There he copied the text of a stele inscribed in Greek and known today as the *Monumentum Adulitanum I*. The text celebrates the military conquests of Ptolemy III Euergetes (reign: 246–222 BCE) and notes the local availability of war elephants for himself and his predecessor, Ptolemy II Philadelphus (reign: 284–246 BCE) (*Bowersock, 2013*).

Echoing this account is the first-century *Periplus Maris Erythraei*, an anonymous text focused on maritime trade across the Red Sea Basin: 'practically the whole number of elephants and rhinoceros that are killed live in the places inland, although at rare intervals they are hunted on the seacoast even near Adulis' (*Casson, 1989*; *Casson, 1993*). Pliny the Elder described Adulis as a thriving emporium in his *Naturalis Historia*, another first-century text, and commented on the availability of ivory, rhinoceros horn, hippopotamus hides, tortoise shell, and *sphingia*—or 'sphinx monkeys,' a term that probably refers to the gelada, *Theropithecus gelada* (*Jolly and Ucko, 1969*). Pliny's account relied heavily on the writings of Agatharchides of Cnidus (ca. 145 BCE), who described 'Aithiopia' (meaning the Red Sea coast and African hinterlands) as a source of sphinx monkeys, *cepi* (probably patas monkeys, *Erythrocebus patas*; *Burstein, 1989*), and *cynocephali*—or 'dog-heads.' Strabo's *Geographica* references the worship of *cynocephali* at Hermopolis (Egypt), making it clear that the animal in question is the hamadryas baboon (*Papio hamadryas*), the traditional sacred animal of the Egyptian god Thoth (*Figure 1*). The source of baboons in ancient Egypt is an enduring question (*Dominy et al., 2020*) as the current distribution of baboons excludes Egypt (*Figure 2*) and there is no prehistoric evidence of baboons occurring in Egypt naturally (*Geraads, 1987*).

Though fragmentary, this historiography points to Adulis as a commercial source of mummified baboons in Ptolemaic catacombs, such as those at Saqqara and Tuna el-Gebel (*Goudsmit and Brandon-Jones, 1999*; *Peters, 2020*) [or those of their progenitors if Ptolemaic Egyptians maintained captive breeding programs; (*von den Driesch et al., 2004*)]. At the same time, these accounts invite questions focused on the source of pre-Ptolemaic baboons recovered from Gabbanat el-Qurud, Egypt (*Lortet and Gaillard, 1907*) and dated to ca. 800–540 BCE (*Richardin et al., 2017*), a span that corresponds to the 25th Dynasty and Late Period of Egyptian antiquity. If these older specimens can be traced to Eritrea, and by extension Adulis, then they have the potential to extend the time depth of Egyptian–Adulite trade by as much as five centuries.

Mummified baboons have been investigated morphologically, revealing species-level taxonomic assignments as well as individual details, such as age, sex, and pathological condition (*Boessneck, 1987*; *Brandon-Jones and Goudsmit, 2022*; *Goudsmit and Brandon-Jones, 1999*; *Peters, 2020*). Such data are telling, but insufficient for determining fine-scale geographic origins. Recent oxygen and strontium stable isotope evidence suggests that mummified hamadryas baboons were imported from a region encompassing northern Somalia, Eritrea, and Ethiopia (*Dominy et al., 2020*), a level of

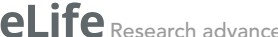

**Figure 1.** Strabo's reference (17.1.40) to the worship of cynocephali at Hermopolis Magna makes clear that the animal in question is the hamadryas baboon (*Papio hamadryas*). The sanctuary and temple complex featured several 35-tonne statues of *P. hamadryas* as the embodiment of Thoth. One of the oldest deities in the Egyptian pantheon, Thoth is best known as a god of writing and wisdom, a lunar deity, and vizier of the gods, but also as a cosmic deity, creator god, and warrior (**Stadler, 2012**). The quartzite statues were erected by Amenhotep III, 18th Dynasty, New Kingdom, 1390–1353 BCE. Photograph by N.J. Dominy.

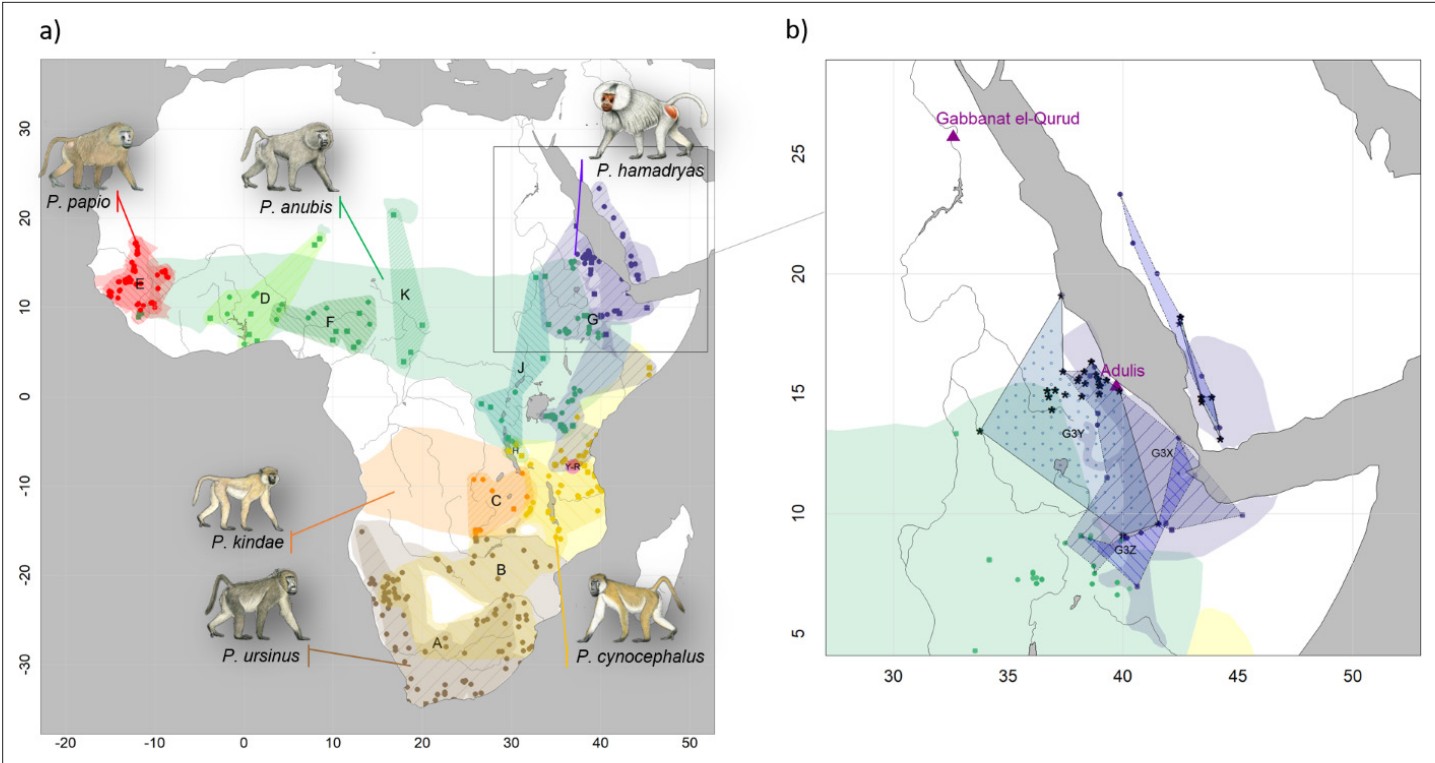

**Figure 2.** Present-day distributions of the six baboon species, major mitochondrial clades, and provenance of samples analysed in this study.
(**a**) Overview of species distributions according to the IUCN (2020) and coloured by species (red: *P. papio*; brown: *P. ursinus*; yellow: *P. cynocephalus*; orange: *P. kindae*; green: *P. anubis*; purple: *P. hamadryas*). Colour-patterned regions reflect main mitochondrial clade attribution resulting from phylogenetic reconstructions and are denoted with capital letters A–K (*Figure 8*). Squares and circles represent geoprovenance of mitogenomes and partial mtDNA datasets (e.g. D-loop, cytochrome *b*), respectively, and are coloured by species. Note that introgressive hybridization has led to discordances between species assignment and mitochondrial clades. (**b**) Detailed view of the distribution of mitochondrial subclades G3-X, G3-Y, and G3-Z in the northeastern distribution of baboons. Samples attributed to G3-Y, the subclade assigned to the mummified baboon in phylogenetic reconstructions and haplotype networks, are highlighted with asterisks. The locations of the excavation site of the mummified baboon, Gabbanat el-Qurud, and Adulis are marked with magenta triangles. Male baboon drawings by Stephen Nash, used with permission.

geographic precision with limited practical value. Another limitation concerns the captive breeding of some animals. For instance, stable isotopes can reveal a lifetime in Egypt but not the geoprovenance of the source population, as shown for olive baboons from the Ptolemaic catacombs of North Saqqara (*Dominy et al., 2020*). The analysis of ancient DNA (aDNA) recovered from baboon mummies and compared to the current distribution of baboon genetic diversity has the potential to provide more detailed insights on the geographic origin of baboons in ancient Egypt. To explore this possibility, we sequenced the mitochondrial genome (mitogenome) of a mummified baboon to infer its geographic origin through phylogenetic assignment.

## Gabbanat el-Qurud

In *Topography of Thebes*, *Wilkinson, 1853* noted a site called Gabbanat el-Qurud ('Valley of the Monkeys') located ca. 2.5 km north–northwest of Medinet Habu, the mortuary temple of Ramses III. Intrigued by this observation, French Egyptologists Louis Lortet and Claude Gaillard sought and found the site in February 1905, along with the remains of mummified baboons. They recovered '17 skulls and a large quantity of bones,' which they attributed to *Papio anubis* and *P. hamadryas* (*Lortet and Gaillard, 1907*). The assemblage includes juvenile and adult males and females buried in jars, sarcophagi, or wooden coffins. Now accessioned in the Musée des Confluences, Lyon, France, the linen wrapping of one mummified individual (MHNL 90001206) was dated radiometrically to 803–544 cal. BC (95.4%) (*Richardin et al., 2017*).

   *Ottoni et al., 2019* sampled dental calculus from 16 individuals in this same assemblage and reported the preservation of ancient microbial DNA in a subset of six. Their success motivated us to

extract DNA from the remaining tooth material of ten individuals (*Table 1*, *Supplementary file 1*). In addition, we obtained samples (skin, bone, or tooth) from 21 modern historic specimens of baboons available in museum collections and representing the northeast African distribution of *Papio* (*Table 1*, *Figure 2*). These specimens were collected between 1855 and 1978, and we denote them 'historic samples' in the remainder of the article to distinguish them both from the older mummified specimens ('mummified samples') and recently collected material ('modern samples'). Latitude–longitude information on the origin of the specimens was either derived from the respective museum database or assigned based on the listed provenance (*Table 1*).

## Results

### Mitogenomes from mummified and historic specimens

We discarded seven historic samples and nine mummified samples from our analysis due to insufficient DNA content, sequencing failure, or low coverage and sequencing depth (*Supplementary file 1*). Thus, our results are based on the newly generated mitogenomes of 14 historic and 1 mummified individual (*Table 1*). In total, we obtained 896,025,770 raw sequence reads, with a mean of 34,462,530 (± SD 27,945,321) raw sequence reads per sample. On average, 95.5% of reads survived trimming and a median of 9934 (range: 244–2,722,354) reads per sample mapped to the reference mitogenome. After removal of duplicates (duplication level median: 25.1%; range: 2.5–92.6%), a median of 7398 (range: 237–497,458) mapped reads per sample resulted in the median final sequencing depth of 26× (range: 0.21–2952×). After exclusion of samples with low quality, the final dataset had a median final sequencing depth of 37× (range: 16–2952×), with a median of 0.4% undetermined sites (range: 0–1.7%) and a median breadth of coverage of at least 3× of 99.3% (range: 97.4–100%) (*Supplementary file 1*). All these metrics differed considerably depending on sample age (historic versus mummified) and DNA concentration (*Figures 3 and 4*). Capture enrichment strongly increased the number of mapped reads and final mean coverage compared to the shotgun approach (*Figures 3 and 4*). GC content of sequences was 40–50% (*Figure 5*) in the same range as the reference genomes.

The sequencing reads of the mummified sample (MHNL51000172) exhibit C to T and G to A misincorporations at 5′ and 3′ ends, reaching frequencies of 3.3 and 1.6% at the first/last position of the read (*Figure 6*). Mapped reads of the mummified sample agreed to median of 99.2% (IQR 1.6%) when focussing on the 125 sites that exhibited fixed differences between subclades and differed at three sites from the variant found in its subclade (*Figure 7a*). When focussing on the 37 sites that are fixed in the subclade of attribution of the mummified baboon but differed in its consensus sequence, mapped reads agreed to a median of 97.3% (IQR 3.1%) (*Figure 7b*).

### Phylogenetic mapping

Phylogenetic trees inferred from maximum likelihood (ML) and Bayesian inference (BI) revealed identical topologies with generally strong node support (100% bootstrap support [BS] and posterior probability [PP] 1.0) and clearly defined geographic clades (*Figure 8*, *Figure 8—figure supplement 1*). These mitochondrial clades did not directly mirror species assignments. Within the northeastern baboons, the central olive baboon clade J from Democratic Republic of the Congo, Tanzania, South Sudan, and southern Sudan diverged first, followed by northern yellow baboons of clade G1 including a sample from Somalia. Hamadryas baboons formed clade G3, which also included olive baboons from the region. Clade G3 contained three subclades: subclade G3-Z comprised hamadryas baboons from Ethiopia and Djibouti; subclade G3-X comprised hamadryas and olive baboons from Ethiopia, Eritrea, and Somalia; and subclade G3-Y comprised hamadryas and olive baboons from northeastern Sudan and Eritrea. The mummified baboon from Gabbanat el-Qurud (MHNL 51000172) was located in subclade G3-Y, closely related to samples from Eritrea and northeastern Sudan.

The median-joining haplotype networks differentiated samples within clade G3 in greater detail and in a more precise geographic context (*Figure 9*, *Figure 9—figure supplement 1*). They revealed the same three subclades within the G3 clade. The HVRI and the cyt *b* networks were concordant both with each other and with the phylogenetic reconstructions in the attribution of samples to the different subclades, but exhibited slight discrepancies in the relation of clades to each other and the positioning of samples within the clades. Subclade G3-X contained hamadryas baboons from Ethiopia, Somalia, and Eritrea. Subclade G3-Z contained samples from Ethiopia, Somalia, Djibouti,

**Table 1.** Information on samples analysed in this study.

| Taxon | Origin | Museum ID | Country | Latitude | Longitude | MitoClade | AccNo | Reference |
|---|---|---|---|---|---|---|---|---|
| *P. hamadryas* | MNHN | MO-1972–357 | ETH | 9.320 | 42.119 | G3-X | OQ538080 | This study |
| *P. hamadryas* | SMNS | SMNS-Z-MAM-001034* | ETH | 11.500 | 39.300 | G3-X | OQ538076 | This study |
| *P. hamadryas* | MfN | ZMB_Mam_025647_(2) | ETH | 14.164 | 38.891 | G3-X | OQ538079 | This study |
| *P. hamadryas* | SMNS | SMNS-Z-MAM-000960 | ERI | 15.783 | 38.453 | G3-X | OQ538078 | This study |
| *P. hamadryas* | NHMUK | ZD.1910.10.3.1 | SOM | 9.933 | 45.200 | G3-X | MT279063 | *Roos et al., 2021* |
| *P. hamadryas* | MfN | ZMB_Mam_012808 | ETH | 9.314 | 42.118 | G3-X | OQ538089 | this study |
| *P. anubis* | Wild | | ETH | 8.968 | 38.571 | G3-X | JX946196 | *Zinner et al., 2013* |
| *P. hamadryas* | MfN | ZMB_Mam_042543_(1) | ETH | 9.593 | 41.866 | G3-Z | OQ538084 | this study |
| *P. hamadryas* | MfN | ZMB_Mam_074849 | DJI | 11.589 | 43.129 | G3-Z | OQ538085 | this study |
| *P. hamadryas* | MNHN | MO-1972–359 | ETH | 6.998 | 40.478 | G3-Z | OQ538086 | this study |
| *P. hamadryas* | SMNS | SMNS-Z-MAM-001288 | SDN | 19.110 | 37.327 | G3-Y | OQ538081 | this study |
| *P. hamadryas* | Wild | | ERI | 15.011 | 38.971 | G3-Y | JX946201 | *Zinner et al., 2013* |
| *P. hamadryas* | SMNS | SMNS-Z-MAM-007509[†] | - | - | - | G3-Y | OQ538082 | this study |
| *P. hamadryas* | MHNL | 51000172 | EGY | - | - | G3-Y | OQ538083 | this study |
| *P. anubis* | SMNS | SMNS-Z-MAM-000584 [‡] | SDN | 13.460 | 33.780 | G3-Y | OQ538075 | this study |
| *P. cynocephalus* | Wild | | TNZ | 7.347 | 37.165 | G1 | JX946199 | *Zinner et al., 2013* |
| *P. cynocephalus* | MNHN | ZM-MO-1977-5 | SOM | 3.243 | 45.471 | G1 | OQ538088 | this study |
| *P. anubis* | NHMUK | ZD1929.4.27.2 | COD | 0.800 | 26.633 | J | MT279061 | *Roos et al., 2021* |
| *P. anubis* | NHMUK | ZD1929.4.27.1 | COD | 1.183 | 27.650 | J | MT279062 | *Roos et al., 2021* |
| *P. anubis* | Wild | 19GNM2220916 | TNZ | 4.679 | 29.621 | J | MG787545 | *Roos et al., 2018* |
| *P. anubis* | SMNS | SMNS-Z-MAM-032128 | SSD | 4.281 | 33.555 | J | OQ538087 | this study |
| *P. anubis* | SMNS | SMNS-Z-MAM-000583 | SDN | 13.333 | 32.729 | J | OQ538077 | this study |
| *P. anubis* | MfN | ZMB_Mam_074869 | CMR | 5.533 | 12.317 | F | OQ538071 | Kopp et al. in prep |
| *P. anubis* | Wild | | NGA | 7.317 | 11.583 | F | JX946198 | *Zinner et al., 2013* |
| *P. anubis* | MfN | ZMB_Mam_074887 | CMR | 9.328 | 12.946 | F | OQ538069 | Kopp et al. in prep |
| *P. anubis* | MfN | ZMB_Mam_074885 | NGA | 7.298 | 10.318 | F | OQ538064 | Kopp et al. in prep |
| *P. anubis* | MfN | ZMB_Mam_074883 | CMR | 6.334 | 9.961 | F | OQ538072 | Kopp et al. in prep |
| *P. papio* | Wild | | SEN | 12.883 | 12.767 | E | JX946203 | *Zinner et al., 2013* |
| *P. anubis* | NHMUK | ZD.1947.586 | SLE | 8.917 | 11.817 | E | MT279064 | *Roos et al., 2021* |
| *P. anubis* | MfN | ZMB_Mam_075043 | TGO | 9.260 | 0.781 | D | OQ538066 | Kopp et al. in prep |

*Table 1 continued on next page*

*Table 1 continued*

| Taxon | Origin | Museum ID | Country | Latitude | Longitude | MitoClade | AccNo | Reference |
|---|---|---|---|---|---|---|---|---|
| *P. anubis* | MfN | ZMB_Mam_011198 | TGO | 6.228 | 1.478 | D | OQ538067 | Kopp et al. in prep |
| *P. anubis* | Wild | | CIV | 8.800 | 3.790 | D | JX946197 | *Zinner et al., 2013* |
| *P. anubis* | MfN | ZMB_Mam_007396_(1) | TGO | 6.950 | 0.585 | D | OQ538065 | Kopp et al. in prep |
| *P. anubis* | NHMUK | ZD.1939.1022 | NER | 17.000 | 7.933 | D | MT279065 | *Roos et al., 2021* |
| *P. anubis* | NHMUK | ZD.1939.1020 | NER | 17.683 | 8.483 | D | MT279066 | *Roos et al., 2021* |
| *P. anubis* | MNHN | ZM-MO-1960-476 | TCD | 20.344 | 16.786 | K | MT279067 | *Roos et al., 2021* |
| *P. anubis* | MNHN | MO-1996-2511 | CAF | 3.905 | 17.922 | K | OQ538068 | Kopp et al. in prep |
| *P. anubis* | NHMUK | ZD.1907.7.8.11 | CAF | 8.000 | 20.000 | K | MT279068 | *Roos et al., 2021* |
| *P. anubis* | MNHN | MO-1996-2510 | CAF | 4.966 | 18.701 | K | OQ538070 | Kopp et al. in prep |
| *P.ursinus* | Wild | | ZAF | 24.680 | 30.790 | B | JX946205 | *Zinner et al., 2013* |
| *P. cynocephalus* | Wild | | TNZ | 11.261 | 37.514 | B | JX946200 | *Zinner et al., 2013* |
| *P. kindae* | | | ZMB | 12.591 | 30.252 | C | JX946202 | *Zinner et al., 2013* |
| *P. cynocephalus* | Wild | 04MNM1300916 | TNZ | 6.119 | 29.730 | H | MT279069 | *Roos et al., 2021* |
| *P. ursinus* | Wild | | ZAF | 34.456 | 20.407 | A | JX946204 | *Zinner et al., 2013* |
| *P. cynocephalus* | Wild | 24UNF1150317 | TNZ | 7.815 | 36.895 | | MT279060 | *Roos et al., 2021* |
| *Theropithecus gelada* | | | | | | | FJ785426 | *Hodgson et al., 2009* |

AccNo, GenBank accession number; NHMUK, Natural History Museum, London; MNHN, Muséum National d'Histoire Naturelle, Paris; MfN, Museum für Naturkunde, Berlin; SMNS, State Museum of Natural History Stuttgart; MdC, Musée des Confluences, Lyon

*Mislabelled in museum records as *T. gelada*.

†Unclear provenance 'Somaliland' (not equal to present-day Somaliland).

‡Misidentified provenance 'Abyssinia' as Ethiopia in museum records.

from the southern tip of Eritrea, and the Arabian Peninsula. Subclade G3-Y contained samples from Eritrea, eastern Sudan, the Arabian Peninsula, and the mummified sample MHNL 51000172. Individuals closely related to this mummified baboon in the cyt *b* network were those from Sudan (on the Red Sea coast and in Senaar), Eritrea (between 14.3–16.0N 36.7–39.0E), and the Arabian Peninsula (*Figure 9—figure supplement 1*), and in the HVRI network samples from location 'Bbr' (Barka Bridge, 15.6N 38.0E) in Eritrea (*Figure 9*).

## Discussion

We succeeded in sequencing the mitogenomes of 14 historic baboons from northeastern Africa and a mummified baboon recovered from Gabbanat el-Qurud, presenting the first genetic data of a mummified baboon from ancient Egypt to date. DNA of the mummified baboon shows *post-mortem*

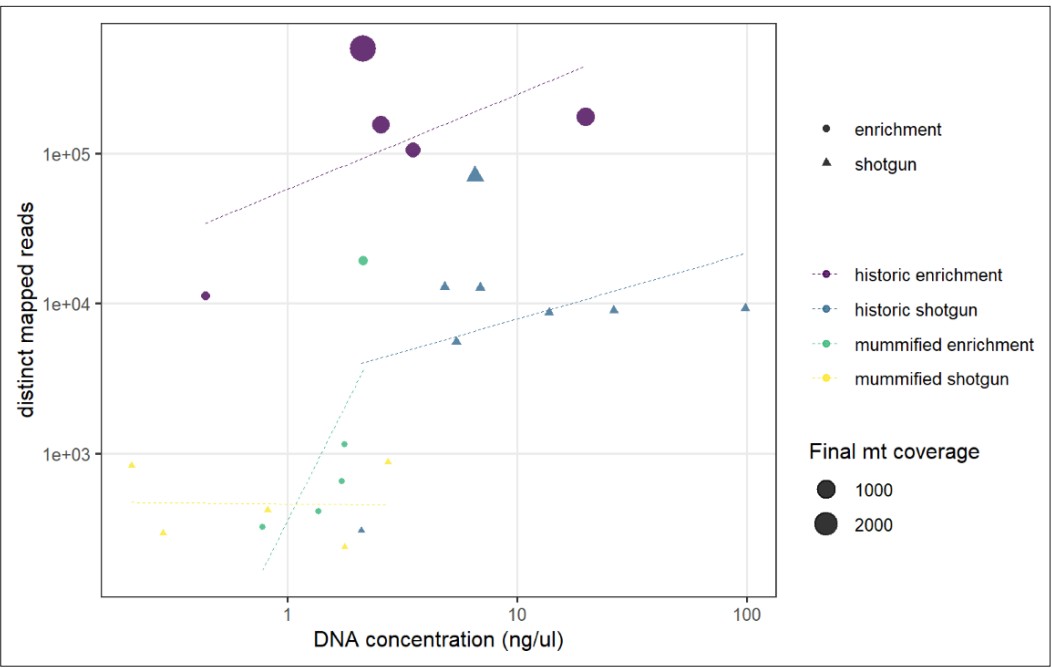

**Figure 3.** Comparison of DNA concentration and amount of distinct mapped reads. A higher DNA concentration produces a higher number of distinct mapped reads. Capture enrichment additionally increases the number of distinct mapped reads. Circles and triangles depict the different sequencing approaches, enrichment, and shotgun, respectively; size is related to the final coverage of the mitogenome; colours represent the different sample types and sequencing approaches (yellow: shotgun sequencing of the mummified sample, MHNL 51000172; blue: shotgun sequencing of historic sample; purple: capture enrichment of historic sample; green: capture enrichment of mummy sample).

damage, which is, however, relatively low compared to what can be expected for samples of similar age (*Dabney et al., 2013b*, *Kistler et al., 2017*). Low frequencies of *post-mortem* damage were observed for aDNA from mummified specimens and have been attributed to the water deprivation during the mummification process, which may prevent hydrolytic deamination (*Rossi et al., 2021*). *Post-mortem* damage observed here is within the range previously reported for aDNA derived from mummified Egyptians (*Schuenemann et al., 2017*) and sheep recovered from an Iranian saltmine (*Rossi et al., 2021*), which supports the authentic origin of our ancient sequence data and tends to rule out the possibility of contamination with modern DNA. The very low frequency of mismatches in the mapped reads from the mummified sample and its unique sequence are further evidence against the concern of contamination from other baboon samples.

Our phylogenetic analysis of the newly generated mitogenomes in combination with published mitochondrial sequence data produced tree topologies in agreement with those of prior studies, with three well-supported clades across the northeastern distribution of *Papio* (*Roos et al., 2021*). As previously described, introgressive hybridization has led to discordances between species assignment and mitochondrial clades (*Rogers et al., 2019*; *Sørensen et al., 2023*; *Zinner et al., 2009*; *Zinner et al., 2011*). Our findings are notable for including specimens from previously unsampled or underrepresented regions, filling gaps in our knowledge of the distribution of mitochondrial clades. For instance, we report mitochondrial sequence data of baboons from regions previously unstudied, namely South Sudan and Sudan. We show that samples from South and southern Sudan, east of the White Nile, nest within the central olive baboon clade J, whereas samples from the coastal region of Sudan and east of the Blue Nile nest within the hamadryas clade G3. These findings expand the northern distributions of both clade J and clade G3 significantly, while also highlighting a strong geographic affinity between clade J and the Albertine Rift and (White) Nile Valley. Taxonomically, this clade corresponds with two subspecies recognized by *Hill, 1970*: *P. a. heuglini* and *P. a. tesselatum*.

A mummified hamadryas baboon from Gabbanat el-Qurud (MHNL 51000172) yielded sufficient aDNA to produce a complete mitogenome, which fell unequivocally in subclade G3-Y (cf. *Kopp*

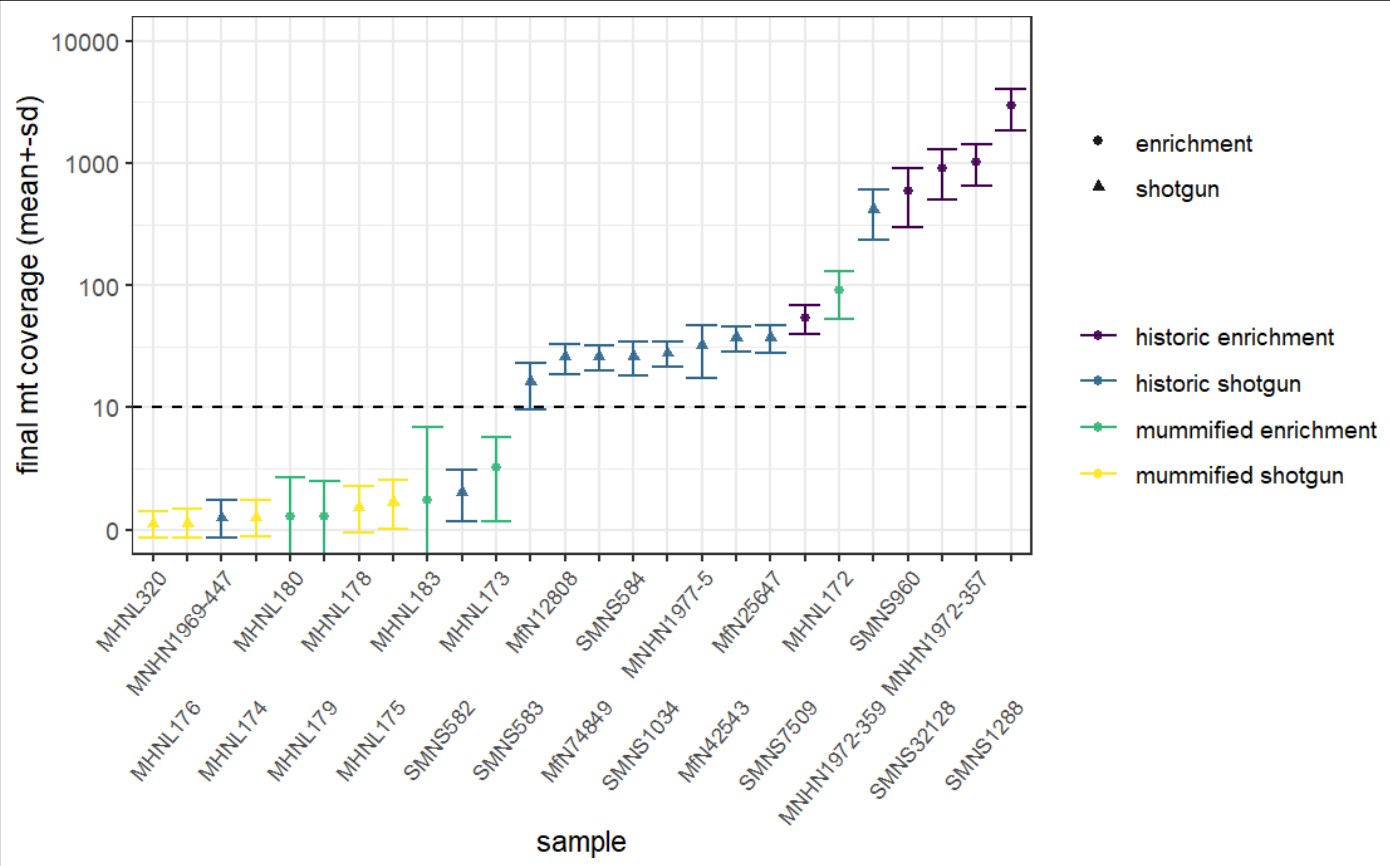

**Figure 4.** Overview of sequencing success for museum and mummy specimens. Mean (± SD) final coverage of the mitogenome is shown for each sample (with abbreviated museum ID). Circles and triangles depict the different sequencing approaches, enrichment and shotgun, respectively; colours represent the different sample types and sequencing approaches (yellow: shotgun sequencing of mummy sample; blue: shotgun sequencing of historic sample; purple: capture enrichment of historic sample; green: capture enrichment of mummy sample). Dashed line shows the cut-off limit 10× for mean final coverage; samples below were excluded from final analyses.

*et al., 2014b*). Haplotype networks allowed us to further refine subclade G3-Y, which consists of *P. hamadryas* and *P. anubis* samples from Eritrea and *P. anubis* samples from neighbouring regions in Sudan. G3-Y also includes samples from the southern-most distribution of *P. hamadryas* on the Arabian Peninsula. Geographic stability of phylogenetic clades over millennia has been shown for other baboon populations (*Mathieson et al., 2020*), leading us to infer that MHNL 51000172 (or its maternal ancestor) originated in the region where clade G3-Y exists today. We cannot completely rule out an Arabian origin for MHNL 51000172, as our data does not cover the entire historic and present haplotype diversity there, but the tight clustering of the currently available Arabian sequences and distances in the HVRI network make an Arabian origin of MHNL 51000172 unlikely. Similarly, the close relationship with a sample of *P. anubis* from Sudan east of the Blue Nile (SMNS-Z-MAM-000584) could indicate trafficking of baboons along the Nile, as suggested for specimens of *P. anubis* recovered from Ptolemaic catacombs (*Brandon-Jones and Goudsmit, 2022*; *Peters, 2020*) and the Predynastic site of Hierakonpolis (*Van Neer et al., 2004*). However, MHNL 51000172 was identified phenotypically as *P. hamadryas* (*Lortet and Gaillard, 1907*), and the distribution of hamadryas baboons is restricted to more eastern regions (*Figure 2*). If the distributions of baboons in northeastern Africa have remained roughly stable within the last 2500 y (as supported by ecological niche modelling; *Chala et al., 2019*), the region in Sudan east of the Blue Nile and west of the Atbarah River could not have served as a source region for hamadryas baboons. Thus, it stands to reason that MHNL 51000172 (or its maternal ancestor) was captured in present-day Eritrea (or close neighbouring regions) and trafficked to Egypt. The value of this finding is twofold. First, it connects the mummified baboon to populations that live today in Eritrea and eastern Sudan, between 13° and 20° latitude. Second, it corroborates the reports

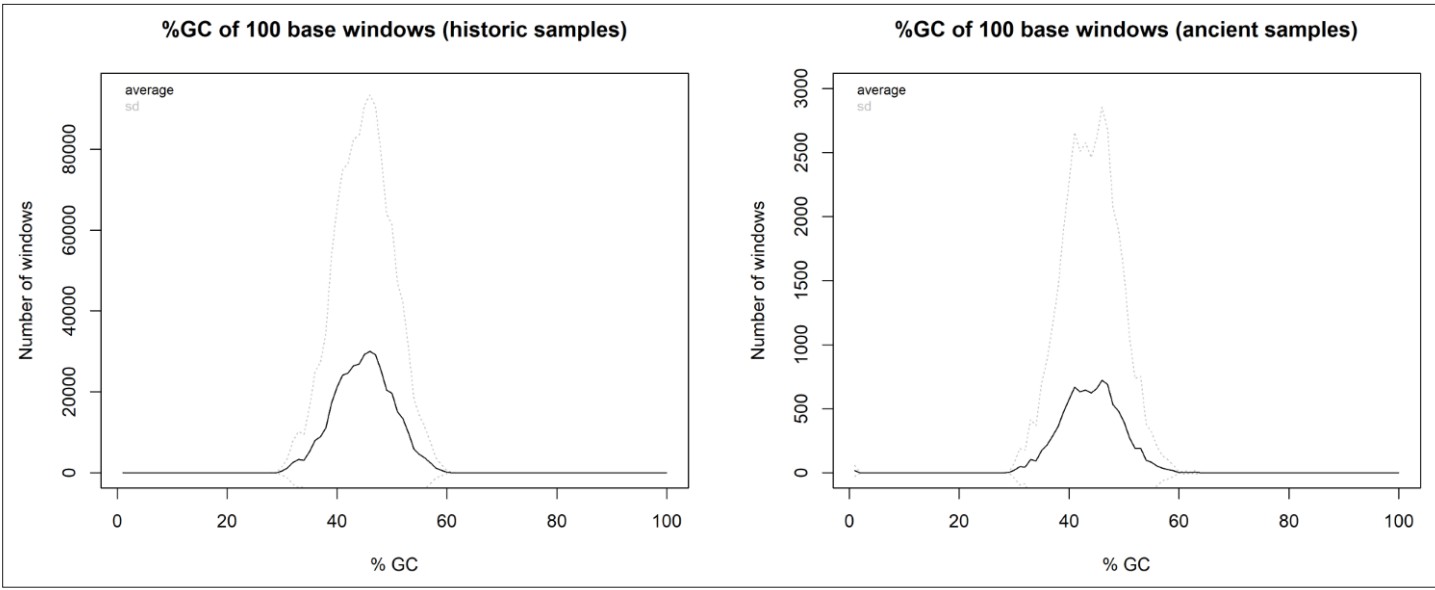

**Figure 5.** Distribution of GC content in historic samples and mummified samples.

of Greco-Roman historians, who described Eritrea, and specifically Adulis, as the sole source of *P. hamadryas* for Ptolemaic Egyptians.

Yet, this baboon predates the reign of Ptolemy I by centuries, presuming it is contemporaneous with another baboon (MHNL 90001206) in the same assemblage, ca. 800–540 BCE (*Richardin et al., 2017*). Thus, our findings raise the possibility that Adulis already existed as a trading centre or entrepôt during the 25th and 26th dynasties of Egypt. Although speculative, and expressed with due caution, our reasoning would extend the antiquity of Egyptian–Adulite trade by as much as five centuries.

Arguing for pre-Ptolemaic contact between Egypt and Adulis is fraught in the absence of corroborating material evidence—but even so, the archaeological record is not entirely silent on the prospect. *Manzo, 2010* and others (*Zazzaro et al., 2014*) reassessed the ceramic tradition at Adulis and developed a chronology that stretches to the early second millennium BCE, the deepest levels of which contained a fragment of blue glass with yellow inlays similar to Egyptian glass from the New

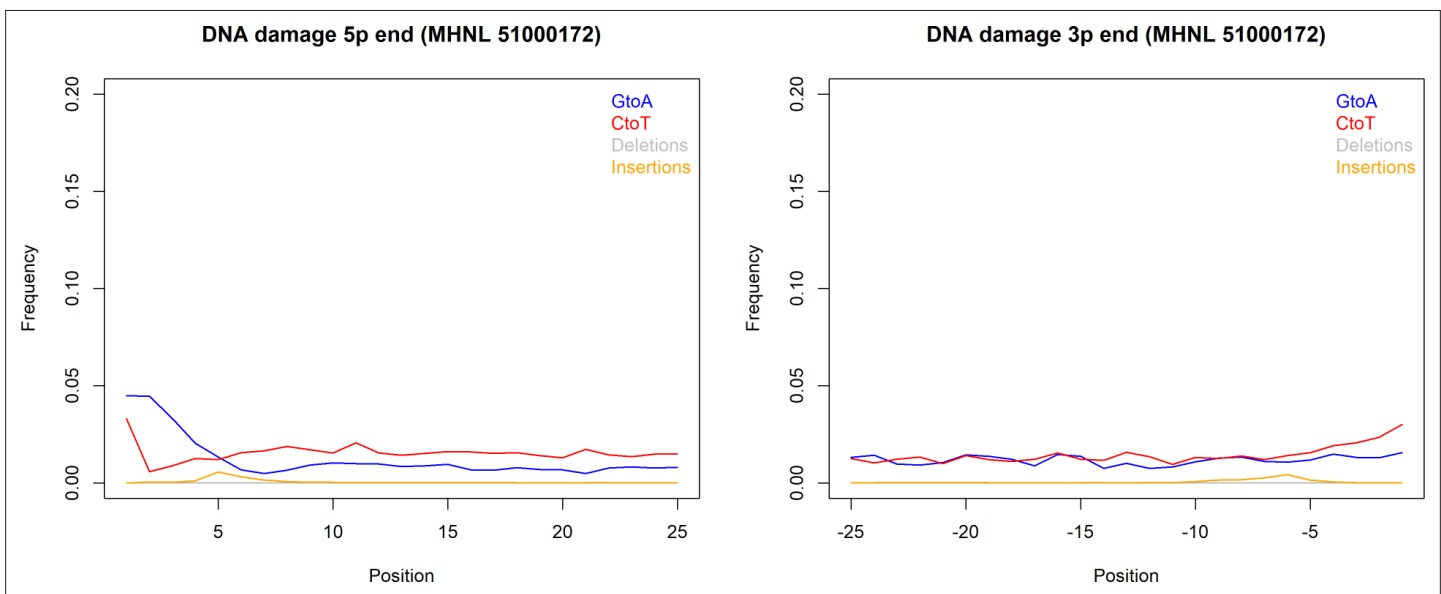

**Figure 6.** DNA damage plot for the sample of the mummified baboon MHNL 51000172 from 5' and 3' read ends, showing mean frequencies of C to T substitutions (red), G to A substitutions (blue), deletions (grey), and insertions (yellow) over the first/last 25 positions.

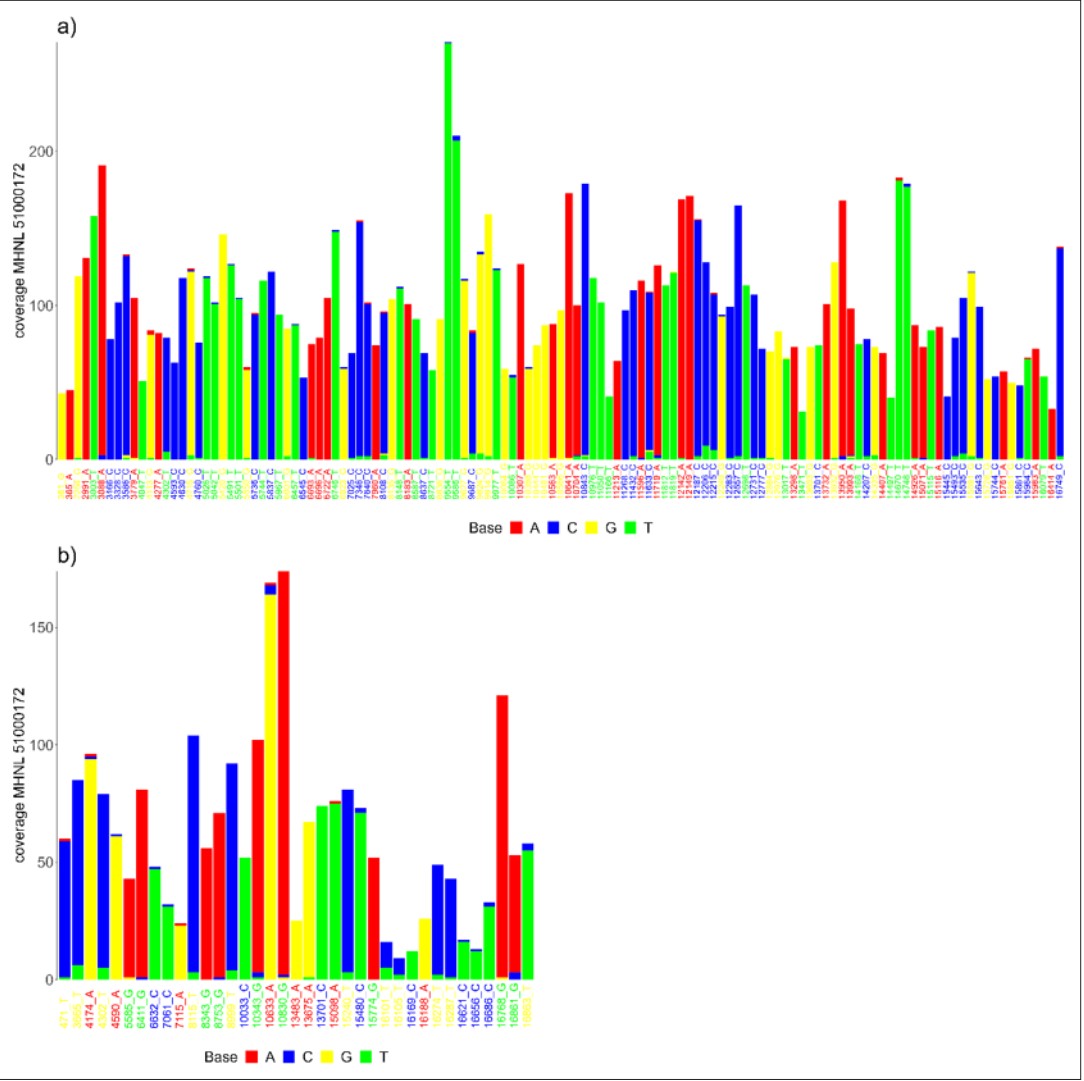

**Figure 7.** Barplots showing the bases of mapped reads for the sample of the mummified baboon MHNL 51000172 at sites that (**a**) exhibit fixed differences among northeastern subclades and (**b**) are fixed in subclade G3-Y but differ in the consensus sequence of the mummified baboon. Sites are named according to their position and the base in the G3-Y consensus sequence and coloured by base. Bases are colour-coded (A: red; C: blue; G: yellow, T: green).

Kingdom (**Fattovich, 2018**). In Egypt, contact with the Eritrean lowlands is attested by trade goods dating to ca. 1800–1650 BCE or earlier, including potsherds, obsidian, and fragments of carbonized ebony (**Fattovich, 2018**; **Lucarini et al., 2020**). Discovered at Mersa Gawasis, a Middle Kingdom harbour, these artefacts appear to align the prehistory of Adulis with the fabled Land of Punt (**Bard and Fattovich, 2018**; **Manzo, 2010**; **Manzo, 2012**), an enigmatic toponym scattered across scant and disconnected records (**Cooper, 2020**).

Punt existed in a region south and east of Egypt, and was accessible by land or sea. For Egyptians, Punt was a source of 'marvels,' particularly incense, but also baboons, that drove bidirectional trade for 1300 y (ca. 2500–1170 BCE) (**Tallet, 2013**). Some scholars have described this enterprise as the beginning of economic globalization (**Fattovich, 2012**), whereas others view it as the earliest maritime leg of the spice route (**Keay, 2006**), a trade network that would shape geopolitical fortunes for millennia. The global historical importance of Punt is therefore considerable, but there is a problem—its location is uncertain, in part because the toponym fades from view. From the early first millennium BCE, there are no further records of Egyptians in Punt or of Puntites visiting Egypt. There are, however, two incomplete inscriptions that mention Punt in a narrative context, and both are attributed to the 26th (Saite) Dynasty (**Betrò, 1996**; **Cavasin, 2019**). One of these, the Defenneh stele, describes

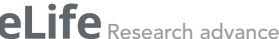

**Figure 8.** Phylogeny of baboons based on complete mitochondrial genomes as inferred from maximum likelihood analysis. *P. cynocephalus* from the Udzungwa Mountains and outgroup *T. gelada* were omitted from visualization for clarity. The analysed baboon mummy sample MHNL 51000172 (in bold) falls into clade G3-Y. Clade names (A–K) according to *Roos et al., 2021*, subclades X–Z according to *Kopp et al., 2014b*; sample IDs include putative species (P.ham, *P. hamadryas*; P.anu, *P. anubis*; P.cyn, *P. cynocephalus*; P.urs, *P. ursinus*; P.pap, *P. papio*), country of origin code (CAF, Central African Republic; CMR, Cameroon; COD, Democratic Republic of Congo; DJI, Djibouti; ERI, Eritrea; ETH, Ethiopia; NGA, Nigeria; SDN, Sudan; SSD, South Sudan; SEN, Senegal; SLE, Sierra Leone; SOM, Somalia; TGO, Togo; note that sample SMNS7509 is of unclear geoprovenance), and abbreviated museum code. Nodes with a branch support below 95% are marked with a grey dot. Mitochondrial genomes generated in this study are marked with an asterisk.

*Figure 8 continued on next page*

*Figure 8 continued*

The online version of this article includes the following figure supplement(s) for figure 8:

**Figure supplement 1.** Phylogeny of baboons based on complete mitochondrial genomes under Bayesian inference.

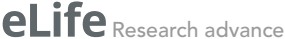

**Figure 9.** Median-joining haplotype network of northeastern baboons based on 644 HVRI sequences (176 bp). The analysed baboon mummy sample resolves in clade G3-Y (depicted in red, black arrow). Circle colour reflects species and country of origin ('Arabia'' comprises samples from Yemen and Saudi Arabia, 'Strait' comprises samples from near the Bab-el-Mandab Strait, i.e. southern Eritrea, Djibouti, northern Somalia).

The online version of this article includes the following figure supplement(s) for figure 9:

**Figure supplement 1.** Median-joining haplotype network of northeastern baboons based on 137 cyt *b* sequences (1140 bp).

an expedition to Punt that was saved from dying thirst by unexpected rainfall on 'the mountains of Punt' (*Meeks, 2003*). The Defenneh stele is a testament to the efforts of Saitic pharaohs to revive maritime commerce on the Red Sea (*Lloyd, 1977*), while also raising the possibility of renewed trade with Punt. It is perhaps no coincidence that the Saite dynasty (664–525 BCE) exists squarely within the radiometric date range of hamadryas baboons from Gabbanat el-Qurud.

Punt, like Adulis, was a source of baboons for Egyptians, a history that raises the possibility of using baboons as a tool for testing geographic hypotheses. Recently, *Dominy et al., 2020* used stable isotope mapping methods to determine the geoprovenance of mummified baboons from Thebes (modern-day Luxor) and dated to the (late) New Kingdom. Their results pointed to present-day Ethiopia, Eritrea, or Djibouti, as well as portions of Somalia, an area that corroborates most scholarly views on the location of Punt (*Breyer, 2016*; *Kitchen, 2004*), but see *Meeks, 2002*; *Meeks, 2003*; *Tallet, 2013*. Here, we used aDNA to show that at least one baboon from the 25th Dynasty or Late Period of Egyptian history—a span that coincides with the last known expeditions to Punt, but predates Greco-Roman accounts of Adulis as a source of baboons—can be traced to Eritrea. Thus, our findings appear to establish primatological continuity between Punt and Adulis. Such a conclusion must be viewed with caution, but it bolsters recurrent conjecture among some historical archaeologists: that Punt and Adulis were essentially the same trading centre from different eras of Egyptian antiquity (*Doresse, 1959*; *Fattovich, 2018*; *Kitchen, 2004*; *Massa, 2021*; *Phillips, 1997*; *Sleeswyk, 1983*).

At minimum, our results reinforce the view that ancient Egyptian mariners travelled great distances to acquire living baboons. A great strength of this conclusion is that it is based on distinct but complementary methods, but of course, the sample size is paltry and limited to *P. hamadryas*, one of two baboon species recovered from Gabbanat el-Qurud. Moving forward, it would be desirable to expand the sample size, examine specimens of *P. anubis* as well as nuclear genomic data for increased precision, and include different time intervals of baboon mummification.

## Future directions

Direct radiocarbon dating of MHNL 51000172 and other baboons from Gabbanat el-Qurud is an urgent priority, in part because doing so would put these specimens into conversation with those from the catacombs of Tuna el-Gebel. The oldest gallery at Tuna el-Gebel, Gallery D, is dated to the 26th Dynasty and contains a single species of baboon: *P. anubis*. Some scholars (*Peters, 2020*; *von den Driesch et al., 2004*) have argued that these olive baboons, as well as *Chlorocebus aethiops* (also found in Gallery D), were sourced from the Sudanese Nile Valley and adjacent areas, which predicts membership in clade G3-Y, although clade J is also plausible. Construction of Gallery C began during the first period of Persian rule in Egypt (524–404 BCE) and continued through the 30th and Ptolemaic dynasties. As every phase of Gallery C contains mummified specimens of both *P. anubis* and *P. hamadryas*, there is rich opportunity to explore diachronic changes in trade routes using phylogeographic methods. Uniform membership in clade G3-Y, for example, would affirm that Late Period Egyptians were sourcing *P. hamadryas* from Eritrea as early as the sixth century BCE. Testing this hypothesis may prove rewarding.

## Materials and methods
### DNA extraction and sequencing

DNA damage and degradation is expected from ancient (mummified) and nineteenth/early twentieth-century museum specimens. We therefore analysed mitochondrial DNA (mtDNA), which is available in higher copy numbers than nuclear DNA and holds greater potential for success when sample quality is poor. We analysed complete mitogenomes because they are effective for reconstructing robust mitochondrial phylogenies of modern baboons and have proven to indicate the geographic origin of the corresponding sample reliably (*Roos et al., 2021*; *Zinner et al., 2013*). Recent advances in sequencing technologies allow the successful sequencing of mitogenomes either with a shotgun sequencing approach or, for samples with very low DNA quality and quantity, with a capture enrichment approach (*Schuenemann et al., 2017*; *Shapiro and Hofreiter, 2012*).

We extracted DNA with a specific column-based method aimed at the recovery of short DNA fragments following established protocols and necessary precautions for the analysis of aDNA (*Dabney et al., 2013a*; *Rohland et al., 2004*; *Roos et al., 2021*). In particular, samples from

mummified specimens were extracted separately and in a dedicated aDNA laboratory to prevent cross-contamination. Concentration of DNA extracts was measured on a Qubit fluorometer (Life Technologies, Singapore) and quality checked on a Bioanalyzer (Agilent, Santa Clara, USA) or Tapestation 2200 (Agilent). All samples were initially sequenced with a shotgun approach. Samples with DNA extract concentrations below 4.5 ng/µl or final mitogenome sequencing depth below 10×, and with enough remaining DNA extract, were enriched for mtDNA with a capture approach.

For the shotgun approach, sequencing libraries were prepared with the NEBNext Ultra II DNA Library Prep Kit (New England BioLabs, Frankfurt, Germany) according to the manufacturer's instructions without prior fragmentation. Library concentration and quality were assessed with the Qubit Fluorometer and Bioanalyzer and molarity was estimated via qPCR with the NEBNext Library Quant Kit (New England BioLabs). Libraries were single indexed with NEBNext Multiplex Oligos (New England BioLabs) with 5–11 PCR cycles and cleaned up with the kit's beads.

For the capture enrichment approach, RNA baits (myBaits custom Kit, Arbor Biosciences, Ann Arbor, USA) were designed for the mitogenome of *P. anubis* East (GenBank Acc. No. JX946196; *Zinner et al., 2013*). We prepared libraries with the Accel-NGS 1S Plus DNA Library Kit and the 1S Plus Dual Indexing Kit (Swift Biosciences, Ann Arbor, USA) according to the manufacturer's instructions for small fragment retention. Hybridization capture was performed with a 48 hr incubation step according to the manufacturer's instructions for highly degraded DNA. After library amplification with 14 PCR cycles, libraries were cleaned with SPRIselect beads (Beckmann Coulter, Krefeld, Germany).

Sequencing was performed with 24 libraries per lane (23 samples + pooled negative control to monitor contamination) on an Illumina HiSeq4000 (50 bp, single-end read) at the NGS Integrative Genomics core unit of the University Medical Center Göttingen, Göttingen, Germany, or on a NovaSeq6000 SP flow cell (100 bp, paired-end read) at the Max Planck Institute for Molecular Genetics, Berlin, Germany. Capture enrichment libraries were reloaded and sequenced a second time to increase the number of reads.

## Mitogenome assembly

Raw sequencing reads were demultiplexed and adapters trimmed at the sequencing facilities. We performed subsequent sequence processing on the central high-performance computing cluster bwForCluster BinAC. We checked read quality with FastQC 0.11.8 (*Andrews, 2010*), trimmed and filtered reads with Trimmomatic 0.39 (*Bolger et al., 2014*) using the settings ILLUMINACLIP:TruSeq3-PE.fa:2:30:10 MINLEN:30 SLIDINGWINDOW:4:20 LEADING:20 TRAILING:20, AVGQUAL:30, and confirmed adequate quality of trimmed reads again with FastQC. Reads were mapped with Burrows Wheeler Aligner (BWA) backtrack 0.7.17 (*Li and Durbin, 2009*) using default settings independently to each of the seven different mitogenomes of representatives of the northern baboon clades (*P. anubis* East JX946196; *P. anubis* Gombe MG787545; *P. anubis* West1 JX946197; *P. anubis* West 2JX946198; *P. cynocephalus* North JX946199; *P. hamadryas* JX946201; *P. papio* JX946203). We chose this approach to avoid biases in downstream analyses introduced through the choice of the reference genome and used the consensus sequence resulting from the best mapping results in downstream analyses. We did not adjust the settings as usually recommended to improve mapping results for aDNA (*Schubert et al., 2012*) but were stringent in mapping and filtering of reads to avoid the inclusion of nuclear mitochondrial DNA segments (NUMTs). Alignments were indexed with SAMtools 1.10 'index' and filtered with 'view' for mapped and (in the case of paired-end data) properly paired reads with a mapping quality of at least MAPQ 30. Library complexity was estimated with the 'EstimateLibraryComplexity' from the Picard Toolkit 2.20.4 (*Broad Institute, 2019*). We merged BAM files of the same samples with 'MergeSamFiles' and removed duplicates with 'MarkDuplicates' from the Picard Toolkit. DNA damage was estimated calculating the frequency of base substitutions, insertions, and deletions at the 5′ and 3′ end, respectively, with DamageProfiler 1.0 (*Neukamm et al., 2021*). We calculated average sequencing depth with SAMtools 1.10 'depth' (*Li et al., 2009*) as the sum of reads covering each position divided by the number of bases in the reference genome, and estimated GC-bias with 'CollectGCBiasMetrics' from the Picard Toolkit. We created a consensus sequence for each sample with the 'doFasta' option in ANGSD (*Korneliussen et al., 2014*) using the base with the highest effective depth (EBD) and setting positions with coverage below 2 to undetermined. We only retained mitogenomes for further analyses for which at least 80% of the sequence were covered at 3×.

We augmented our dataset with published mitogenomes of baboons (*Roos et al., 2021*) and *T. gelada* as outgroup (*Table 1*) and aligned sequences with MUSCLE 3.8.81 (*Edgar, 2004*) as implemented in the package msa 1.28.0 (*Bodenhofer et al., 2015*) in R 4.2.1 (*R Development Core Team, 2022*) using standard settings with a maximum number of 16 iterations.

For a more fine-scale geographic representation, we further included published sequence data from the northeastern part of the baboon distribution of two different mitochondrial markers with differing resolution: the cytochrome *b* gene (cyt *b*) (*Zinner et al., 2009*; *Zinner et al., 2015*) and a fragment of the hypervariable region I (HVRI) of the D-loop (*Hapke et al., 2001*; *Kopp et al., 2014a*; *Kopp et al., 2014b*; *Städele et al., 2015*; *Winney et al., 2004*). We extracted the corresponding regions from the mitogenome alignment and again removed sequences with more than 10% undetermined sites.

We assessed contamination by checking mismatches of the mapped reads from the mummified sample at sites in the mitogenome that (i) are distinct between northeastern subclades (125 fixed differences) and (ii) are fixed in subclade G3-Y (considering all samples but the mummified baboon) but differ in the consensus sequence of the mummified sample (37 sites).

## Phylogenetic reconstruction

To identify the phylogenetic affiliation of the newly investigated samples, we reconstructed phylogenetic trees based on the final dataset of 46 mitogenomes (alignment length: 16,628 bp) using ML and BI methods with W-IQ-Tree 1.6.12 (*Nguyen et al., 2015*; *Trifinopoulos et al., 2016*) and MrBayes 3.2.7 (*Huelsenbeck and Ronquist, 2001*; *Ronquist and Huelsenbeck, 2003*), respectively. We treated the mitogenome as a single partition, the optimal substitution model for phylogenetic reconstructions was detected to be TN + F + I + G4 (*Tamura and Nei, 1993*) under the Bayesian information criterion and GTR + F + I + G4 (*Tavaré, 1986*) under the Corrected Akaike Information Criterion with Modelfinder (*Kalyaanamoorthy et al., 2017*) as implemented in W-IQ-Tree. The ML tree was reconstructed with 10,000 ultrafast bootstrap replications (*Hoang et al., 2018*) applying the TN + F + I + G4 model. The BI tree was reconstructed applying the GTR + I + G model and using four independent Markov chain Monte Carlo runs with 1 million generations, a burn-in of 25%, and sampling every 100 generations. To ensure convergence, the Potential Scale Reduction Factor was checked to be close to 1 for all parameters. We visualized phylogenetic trees with the R package ggtree 3.4.2 (*Yu et al., 2017*) and adopted clade nomination of *Roos et al., 2021* and *Kopp et al., 2014b*.

## Haplotype networks

To determine the mitochondrial clade of origin of the analysed samples more precisely, we reconstructed median-joining haplotype networks (*Bandelt et al., 1999*) with Popart 1.7 (*Leigh and Bryant, 2015*) for both the HVRI (n = 644, 176 bp) and the cyt *b* (n = 137, 1140 bp) dataset.

## Geographic maps

Geographic maps were created in R. We obtained species distribution shapefiles from IUCN (*Gippoliti, 2019*; *Sithaldeen, 2019*; *Wallis, 2020a*; *Wallis, 2020b*; *Wallis et al., 2020*; *Wallis et al., 2021*), river, lake and coastlines from Natural Earth (https://www.naturalearthdata.com) via rnaturalearth 0.1.0 (*Massicotte and South, 2023*).

## Acknowledgements

We thank Frieder Mayer and Christiane Funk from MfN Berlin for sharing baboon samples with us for genetic analyses. We thank Christiane Schwarz for assistance in DNA extraction and library preparation and Bernd Timmermann and Stefan Börno for advice and facilitation of sequencing. We are grateful to Salima Ikram and Julien Cooper for energizing discussion on the topics of Egyptian mummification and toponyms, and thank Laura Epp for advice on aDNA analyses. Research carried out on the mummies curated at the Musée des Confluences (Lyon, France) is supported by the SIMoS Program funded by LabEx ARCHIMEDE from 'Investir L'Avenir' program ANR-11-LABX-0032-01 to SP. We acknowledge the Service des Musées de France, Mme Dominique Dupuis-Labbé and the Ministère de la Culture et de la Communication (France) for their ongoing support to the research carried out on the mummies. We acknowledge the support by the High Performance and Cloud Computing Group at the Zentrum für Datenverarbeitung of the University of Tübingen, the state of

Baden-Württemberg through bwHPC and the German Research Foundation (DFG) through grant no INST 37/935-1 FUGG. We acknowledge the University of Konstanz Sequencing Analysis (SequAna) Core Facility for bioinformatic assistance. This study was funded by the Young Scholar Fund and the Zukunftskolleg, University of Konstanz (funded by the Federal Ministry of Education and Research (BMBF) and the Baden-Württemberg Ministry of Science as part of the Excellence Strategy of the German Federal and State Governments), and the Junge Akademie at the Berlin-Brandenburg Academy of Sciences and Humanities and the German National Academy of Sciences Leopoldina. NJD received support through a Senior Fellowship at the Zukunftskolleg, GHK was supported by the Hector Pioneer Fellowship of Hector Stiftung II and the Zukunftskolleg. Some of the views in this paper were developed during a workshop titled, "Animating Ancient Trade Routes Through Primate Lifeways," funded by the Wenner-Gren Foundation and Zukunftskolleg. Finally, we thank the handling editors, Julien Cooper, and an anonymous reviewer for their thoughtful suggestions.

## Additional information

### Funding

| Funder | Grant reference number | Author |
|---|---|---|
| Universität Konstanz | Young Scholar Fund | Gisela H Kopp |
| Universität Konstanz | Zukunftskolleg | Nathaniel J Dominy Gisela H Kopp |
| Max-Planck-Gesellschaft | Open Access Fund | Gisela H Kopp |
| Hector Stiftung II | Hector Pioneer Fellowship | Gisela H Kopp |
| Deutsche Akademie der Naturforscher Leopoldina - Nationale Akademie der Wissenschaften | Die Junge Akademie | Gisela H Kopp |
| Bundesministerium für Bildung und Forschung | Excellence Strategy of the German Federal and State Governments | Benjamin Hume Gisela H Kopp |
| Ministerium für Wissenschaft, Forschung und Kunst Baden-Württemberg | bwHPC | Franziska Grathwol Benjamin Hume Gisela H Kopp |
| Deutsche Forschungsgemeinschaft | INST 37/935- 1 FUGG | Franziska Grathwol Benjamin Hume Gisela H Kopp |
| Agence Nationale de la Recherche | ANR-11-LABX-0032-01 | Stéphanie M Porcier |
| Deutsche Forschungsgemeinschaft | Centre of Excellence 2117 "Centre for the Advanced Study of Collective Behaviour" ID: 422037984 | Gisela H Kopp |

The funders had no role in study design, data collection and interpretation, or the decision to submit the work for publication. Open access funding provided by Max Planck Society.

### Author contributions

Franziska Grathwol, Data curation, Formal analysis, Investigation, Visualization, Methodology, Writing - original draft, Project administration; Christian Roos, Resources, Data curation, Supervision, Methodology, Writing - review and editing; Dietmar Zinner, Data curation, Writing - review and editing; Benjamin Hume, Software, Formal analysis, Writing - review and editing; Stéphanie M Porcier, Didier Berthet, Jacques Cuisin, Stefan Merker, Resources, Data curation, Writing - review and editing;

Claudio Ottoni, Resources, Formal analysis, Methodology, Writing - review and editing; Wim Van Neer, Writing - review and editing; Nathaniel J Dominy, Conceptualization, Investigation, Writing - review and editing; Gisela H Kopp, Conceptualization, Resources, Data curation, Software, Formal analysis, Supervision, Funding acquisition, Investigation, Visualization, Methodology, Writing - original draft, Project administration, Writing - review and editing

### Author ORCIDs
Christian Roos  http://orcid.org/0000-0003-0190-4266
Dietmar Zinner  http://orcid.org/0000-0003-3967-8014
Wim Van Neer  http://orcid.org/0000-0003-1710-3623
Nathaniel J Dominy  http://orcid.org/0000-0001-5916-418X
Gisela H Kopp  http://orcid.org/0000-0001-8396-3264

### Decision letter and Author response
Decision letter https://doi.org/10.7554/eLife.87513.sa1
Author response https://doi.org/10.7554/eLife.87513.sa2

---

# Additional files

### Supplementary files
• Supplementary file 1. Overview of analysed samples and sequencing results (provided as .csv).
• MDAR checklist

### Data availability
Raw sequencing data are deposited in the European Nucleotide Archive (ENA, project accession no. PRJEB60261), mitochondrial genomes on Genbank (accession numbers: OQ538075-OQ538089). Code used for data processing and analysis is available on OSF via https://doi.org/10.17605/OSF.IO/D5GX3.

The following datasets were generated:

| Author(s) | Year | Dataset title | Dataset URL | Database and Identifier |
|---|---|---|---|---|
| Kopp GH | 2023 | Adulis and the transshipment of baboons during classical antiquity | https://doi.org/10.17605/OSF.IO/D5GX3 | Open Science Framework, 10.17605/OSF.IO/D5GX3 |
| Kopp GH | 2023 | Adulis and the transshipment of baboons during classical antiquity | https://www.ebi.ac.uk/ena/browser/view/PRJEB60261 | European Nucleotide Archive, PRJEB60261 |
| Grathwol F, Roos C, Zinner D, Hume B, Porcier SM, Berthet D, Cuisin J, Merker S, Ottoni C, Van Neer W, Dominy NJ, Kopp GH | 2023 | *Papio anubis* isolate mitoclade G3-Y voucher Z-MAM-000584 mitochondrion, complete genome | https://www.ncbi.nlm.nih.gov/nuccore/OQ538075 | NCBI GenBank, OQ538075 |
| Grathwol F, Roos C, Zinner D, Hume B, Porcier SM, Berthet D, Cuisin J, Merker S, Ottoni C, Van Neer W, Dominy NJ, Kopp GH | 2023 | *Papio hamadryas* isolate mitoclade G3-X voucher Z-MAM-001034 mitochondrion, complete genome | https://www.ncbi.nlm.nih.gov/nuccore/OQ538076 | NCBI GenBank, OQ538076 |
| Grathwol F, Roos C, Zinner D, Hume B, Porcier SM, Berthet D, Cuisin J, Merker S, Ottoni C, Van Neer W, Dominy NJ, Kopp GH | 2023 | *Papio anubis* isolate mitoclade J voucher Z-MAM-000583 mitochondrion, complete genome | https://www.ncbi.nlm.nih.gov/nuccore/OQ538077 | NCBI GenBank, OQ538077 |

*Continued on next page*

*Continued*

| Author(s) | Year | Dataset title | Dataset URL | Database and Identifier |
|---|---|---|---|---|
| Grathwol F, Roos C, Zinner D, Hume B, Porcier SM, Berthet D, Cuisin J, Merker S, Ottoni C, Van Neer Z, Dominy NJ, Kopp GH | 2023 | *Papio hamadryas* isolate mitoclade G3-X voucher Z-MAM-000960 mitochondrion, complete genome | https://www.ncbi.nlm.nih.gov/nuccore/OQ538078 | NCBI GenBank, OQ538078 |
| Grathwol F, Roos C, Zinner D, Hume B, Porcier SM, Berthet D, Cuisin J, Merker S, Ottoni C, Van Neer W, Dominy NJ, Kopp GH | 2023 | *Papio hamadryas* isolate mitoclade G3-X voucher ZMB_Mam-025647 mitochondrion, complete genome | https://www.ncbi.nlm.nih.gov/nuccore/OQ538079 | NCBI GenBank, OQ538079 |
| Grathwol F, Roos C, Zinner D, Hume B, Porcier SM, Berthet D, Cuisin J, Merker S, Ottoni C, Van Neer W, Dominy NJ, Kopp GH | 2023 | *Papio hamadryas* isolate mitoclade G3-X voucher MO-1972-357 mitochondrion, complete genome | https://www.ncbi.nlm.nih.gov/nuccore/OQ538080 | NCBI GenBank, OQ538080 |
| Grathwol F, Roos C, Zinner D, Hume B, Porcier SM, Berthet D, Cuisin J, Merker S, Ottoni C, Van Neer W, Dominy NJ, Kopp GH | 2023 | *Papio hamadryas* isolate mitoclade G3-Y voucher Z-MAM-001288 mitochondrion, complete genome | https://www.ncbi.nlm.nih.gov/nuccore/OQ538081 | NCBI GenBank, OQ538081 |
| Grathwol F, Roos C, Zinner D, Hume B, Porcier SM, Berthet D, Cuisin J, Merker S, Ottoni C, Van Neer W, Dominy NJ, Kopp GH | 2023 | *Papio hamadryas* isolate mitoclade G3-Y voucher Z-MAM-007509 mitochondrion, complete genome | https://www.ncbi.nlm.nih.gov/nuccore/OQ538082 | NCBI GenBank, OQ538082 |
| Grathwol F, Roos C, Zinner D, Hume B, Porcier SM, Berthet D, Cuisin J, Merker S, Ottoni C, Van Neer W, Dominy NJ, Kopp GH | 2023 | *Papio hamadryas* voucher 51000172 mitochondrion, complete genome | https://www.ncbi.nlm.nih.gov/nuccore/OQ538083 | NCBI GenBank, OQ538083 |
| Grathwol F, Roos C, Zinner D, Hume B, Porcier SM, Berthet D, Cuisin J, Merker S, Ottoni C, Van Neer W, Dominy NJ, Kopp GH | 2023 | *Papio hamadryas* isolate mitoclade G3-Z voucher ZMB_Mam-042543 mitochondrion, complete genome | https://www.ncbi.nlm.nih.gov/nuccore/OQ538084 | NCBI GenBank, OQ538084 |
| Grathwol F, Roos C, Zinner D, Hume B, Porcier SM, Berthet D, Cuisin J, Merker S, Ottoni C, Van Neer W, Dominy NJ, Kopp GH | 2023 | *Papio hamadryas* isolate mitoclade G3-Z voucher ZMB_Mam-074849 mitochondrion, complete genome | https://www.ncbi.nlm.nih.gov/nuccore/OQ538085 | NCBI GenBank, OQ538085 |

*Continued*

| Author(s) | Year | Dataset title | Dataset URL | Database and Identifier |
|---|---|---|---|---|
| Grathwol F, Roos C, Zinner D, Hume B, Porcier SM, Berthet D, Cuisin J, Merker S, Ottoni C, Van Neer W, Dominy NJ, Kopp GH | 2023 | *Papio hamadryas* isolate mitoclade G3-Z voucher MO-1972-359 mitochondrion, complete genome | https://www.ncbi.nlm.nih.gov/nuccore/OQ538086 | NCBI GenBank, OQ538086 |
| Grathwol F, Roos C, Zinner D, Hume B, Porcier SM, Berthet D, Cuisin J, Merker S, Ottoni C, Van Neer W, Dominy NJ, Kopp GH | 2023 | *Papio anubis* isolate mitoclade J voucher Z-MAM-032128 mitochondrion, complete genome | https://www.ncbi.nlm.nih.gov/nuccore/OQ538087 | NCBI GenBank, OQ538087 |
| Grathwol F, Roos C, Zinner D, Hume B, Porcier SM, Berthet D, Cuisin J, Merker S, Ottoni C, Van Neer W, Dominy NJ, Kopp GH | 2023 | *Papio cynocephalus* voucher ZM-MO-1977-5 mitochondrion, partial genome | https://www.ncbi.nlm.nih.gov/nuccore/OQ538088 | NCBI GenBank, OQ538088 |
| Grathwol F, Roos C, Zinner D, Hume B, Porcier SM, Berthet D, Cuisin J, Merker S, Ottoni C, Van Neer W, Dominy NJ, Kopp GH | 2023 | *Papio hamadryas* isolate mitoclade G3-X voucher ZMB_Mam_012808 mitochondrion, complete genome | https://www.ncbi.nlm.nih.gov/nuccore/OQ538089 | NCBI GenBank, OQ538089 |

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
