## [Editor Report]

This fundamental Research Advance sheds new light on the ancient baboon trade in the Red Sea. Combining ancient DNA methods from a mummified baboon with historical accounts, this work provides compelling evidence connecting the ancient Egyptian trade of baboons with the ancient port city of Adulis. The results will be of significance to a broad range of scholars interested in applying ancient DNA to improve our knowledge of historical events.

---

## [Decision Letter]

**Decision letter after peer review:**

Thank you for submitting your article "Adulis and the transshipment of baboons during classical antiquity" for consideration by *eLife*. Your article has been reviewed by 2 peer reviewers, and the evaluation has been overseen by a Reviewing Editor and Christian Rutz as the Senior Editor. The following individuals involved in review of your submission have agreed to reveal their identity: Julien Cooper (Reviewer #1).

Essential revisions:

The main points for revision concern:

1) clarifying the geographic range attributed in the abstract (and throughout the article) for the provenance of the mummified baboon. Please check/ clarify that Adulis port is inferred from historical accounts and location is more generally coastal Eritrea/eastern Sudan.

2) provide DNA contamination estimates based on the mismatch between reads and the consensus sequence for the historical baboon samples as well for the mummified one. It was also recommended that contamination estimates be restricted to transversions for the mummified baboon. (see Reviewer #2's comments for further clarification).

*Reviewer #1 (Recommendations for the authors):*

I congratulate the authors on this study, this is a solid argument for connecting the Egyptian baboon trade with Eritrea and the authors are able to link this convincingly with the wider Punt debate. The language is carefully worded and provides an accurate assessment of the significance of their findings.

At the outset, it should be noted that as an Egyptologist I am not in a position to comment on the genetic analysis in this paper.

l. 62: small points. The word is Geʽez with the apostrophe facing the other direction (this is a different sound). The other point is that the Monumentum Adulitanum was written in Greek and I know of no scholar that suggests it was written in Ge'ez. There Aksumite stele like this one which have parallel Greek and Ge'ez scripts, but Cosmas did not copy any Ge'ez text nor mention one in his account as far as I am aware. One may have existed but I would just mention the Greek in this text as a Ge'ez is unknowable.

L. 126: Usually put the percentage after the date range for calibrated dates.

L. 408. "Gawasis".

L. 409: perhaps change slightly as what you describe below shows that Egyptians did go to Punt. i.e. "there are only scant and disconnected records of Egyptians in Punt" or alike.

L. 436: While the Defenneh Stele is the last expedition text to mention Punt, there are plenty of references to Punt in Greco-Roman religious texts. Perhaps say "span that coincides with the last known expeditionary records for Punt voyages" (or alike).

L. 442: More of an optional thing to mention; a few scholars (Conti Rossini; de Romanis) have tried to connect one of the Puntite toponyms in the topographical lists with Adulis, but the phonetics for this equation are far from certain and probably incorrect, see Cooper, Toponymy on the Periphery, 346.

*Reviewer #2 (Recommendations for the authors):*

1) My only real technical suggestion is that at least for the mummified baboon, the authors assess and report contamination by looking at mismatch in the mapped mitochondrial reads (particularly at sites known to vary between clades). The concern would be contamination from other baboon samples during either archaeological handling or in the lab.

2) I suggest making it clearer in the abstract that the link to Adulis is quite speculative and that the possible geographic area is not just Eritria but extends well into present-day Sudan and Ethiopia.

3) If I understand correctly the authors do have nuclear data from the shotgun sequencing. I understand it's very low coverage and they might not want to analyze it, but I just want to make sure that they are releasing it freely in case others want to.

---

## [Author Response]

Essential revisions:The main points for revision concern:1) clarifying the geographic range attributed in the abstract (and throughout the article) for the provenance of the mummified baboon. Please check/ clarify that Adulis port is inferred from historical accounts and location is more generally coastal Eritrea/eastern Sudan.2) provide DNA contamination estimates based on the mismatch between reads and the consensus sequence for the historical baboon samples as well for the mummified one. It was also recommended that contamination estimates be restricted to transversions for the mummified baboon. (see Reviewer #2's comments for further clarification).

We thank the editors and reviewers for their thoughtful comments and suggestions, we have addressed all recommendations to improve our manuscript. We also slightly changed the structure of the manuscript following the comments of the editorial support. In particular, we have (1) edited the text to more clearly mirror the uncertainty in the geographical assignment of the mummified baboon (#Rev2.2) and (2) included an additional analysis to estimate contamination in the mummified sample based on mismatches between reads and the consensus sequence in comparison to the historical samples (#Rev2.1). Please find our detailed responses to the reviewers’ recommendations below.

Reviewer #1 (Recommendations for the authors):I congratulate the authors on this study, this is a solid argument for connecting the Egyptian baboon trade with Eritrea and the authors are able to link this convincingly with the wider Punt debate. The language is carefully worded and provides an accurate assessment of the significance of their findings.At the outset, it should be noted that as an Egyptologist I am not in a position to comment on the genetic analysis in this paper.l. 62: small points. The word is Geʽez with the apostrophe facing the other direction (this is a different sound). The other point is that the Monumentum Adulitanum was written in Greek and I know of no scholar that suggests it was written in Ge'ez. There Aksumite stele like this one which have parallel Greek and Ge'ez scripts, but Cosmas did not copy any Ge'ez text nor mention one in his account as far as I am aware. One may have existed but I would just mention the Greek in this text as a Ge'ez is unknowable.

We thank Dr. Cooper for catching this error. It has been corrected. (L.65)

L. 126: Usually put the percentage after the date range for calibrated dates.

We thank Dr. Cooper for catching this lapse. The sentence has been edited to the following:

“L. 125-127: Now accessioned in the Musée des Confluences, Lyon, France, the linen wrapping of one mummified individual (MHNL 90001206) was dated radiometrically to 803-544 cal. BC (95.4%) (Richardin et al., 2017).”

L. 408. "Gawasis".

We thank Dr. Cooper for catching this typographical error. It has been corrected (L.467).

L. 409: perhaps change slightly as what you describe below shows that Egyptians did go to Punt. i.e. "there are only scant and disconnected records of Egyptians in Punt" or alike.

Agreed, the sentence has been edited to the following:

“L.467-470: Discovered at Mersa Gawasis, a Middle Kingdom harbour, these artifacts appear to align the prehistory of Adulis with the fabled Land of Punt (Bard & Fattovich, 2018; Manzo, 2010, 2012), an enigmatic toponym scattered across scant and disconnected records (Cooper, 2020).”

L. 436: While the Defenneh Stele is the last expedition text to mention Punt, there are plenty of references to Punt in Greco-Roman religious texts. Perhaps say "span that coincides with the last known expeditionary records for Punt voyages" (or alike).

Agreed, the sentence has been edited to the following:

“L.493-495. Here, we used aDNA to show that at least one baboon from the 25^th^ Dynasty or Late Period of Egyptian history––a span that coincides with the last known expeditions to Punt, but predates Greco-Roman accounts of Adulis as a source of baboons––can be traced to Eritrea.”

L. 442: More of an optional thing to mention; a few scholars (Conti Rossini; de Romanis) have tried to connect one of the Puntite toponyms in the topographical lists with Adulis, but the phonetics for this equation are far from certain and probably incorrect, see Cooper, Toponymy on the Periphery, 346.

We appreciate this suggestion, and we agree with Cooper’s assessment, but we would rather not wade into this tangential topic.

Reviewer #2 (Recommendations for the authors):1) My only real technical suggestion is that at least for the mummified baboon, the authors assess and report contamination by looking at mismatch in the mapped mitochondrial reads (particularly at sites known to vary between clades). The concern would be contamination from other baboon samples during either archaeological handling or in the lab.

We thank you for this suggestion and added the assessment. We included a figure, which shows the (mis)matches of the mapped reads from the mummified baboon at sites in the mitogenome that (i) are distinct (fixed differences) between north-eastern subclades (125 sites), and (ii) are fixed in subclade G3-Y (when considering all samples but the mummified baboon) but differ in the consensus sequence of the mummified baboon (37 sites). These results show that mapped reads agree to a very large degree ((i) median 99.2%, IQR 1.6%; (ii) median 97.3%, IQR 3.1%) and there is no evidence for significant amounts of contamination. We changed the respective parts in the methods, results, discussion, and supplement accordingly:

Methods L. 604-607: “We assessed contamination by checking mismatches of the mapped reads from the mummified sample at sites in the mitogenome that (i) are distinct between north-eastern subclades (125 fixed differences), and (ii) are fixed in subclade G3-Y (considering all samples but the mummified baboon) but differ in the consensus sequence of the mummified sample (37 sites).”

Results L. 327-342: “Mapped reads of the mummified sample agreed to median of 99.2% (IQR 1.6%) when focussing on the 125 sites that exhibited fixed differences between subclades and differed at three sites from the variant found in its subclade (Figure 7a). When focussing on the 37 sites that are fixed in the subclade of attribution of the mummified baboon but differed in its consensus sequence, mapped reads agreed to a median of 97.3% (IQR 3.1%) (Figure 7b).”

Discussion L411-413: “The very low frequency of mismatches in the mapped reads from the mummified sample and its unique sequence are further evidence against the concern of contamination from other baboon samples.”

2) I suggest making it clearer in the abstract that the link to Adulis is quite speculative and that the possible geographic area is not just Eritria but extends well into present-day Sudan and Ethiopia.

We agree, and we have edited the abstract carefully, including Ethiopia and using the words “corroborates” and “pointing toward” to express uncertainty:

“L. 42-46: Phylogenetic assignment connects the mummified baboon to modern populations of *Papio hamadryas* in Eritrea, Ethiopia, and eastern Sudan. This result, assuming geographical stability of phylogenetic clades, corroborates Greco-Roman historiographies by pointing toward present-day Eritrea, and by extension Adulis, as a source of baboons for Late Period Egyptians.”

3) If I understand correctly the authors do have nuclear data from the shotgun sequencing. I understand it's very low coverage and they might not want to analyze it, but I just want to make sure that they are releasing it freely in case others want to.

We agree, and we have made all raw data (including shotgun sequencing reads) openly available in public repositories: European Nucleotide Archive (ENA), project accession no. PRJEB60261. See L.638-642 “Data access”.